# Construction of N-CDs and Calcein-Based Ratiometric Fluorescent Sensor for Rapid Detection of Arginine and Acetaminophen

**DOI:** 10.3390/nano12060976

**Published:** 2022-03-16

**Authors:** Haiyan Qi, Qiuying Li, Jing Jing, Tao Jing, Chuntong Liu, Lixin Qiu, Rokayya Sami, Mahmoud Helal, Khadiga Ahmed Ismail, Amani H. Aljahani

**Affiliations:** 1College of Chemistry and Chemical Engineering, Qiqihar University, No. 42, Wenhua Street, Qiqihar 161006, China; lqy6212161@163.com (Q.L.); jtkr@163.com (T.J.); lct2291@163.com (C.L.); Qlx980124@163.com (L.Q.); 2School of Medicine and Health, Harbin Institute of Technology, No.92, West Dazhi Street, Harbin 150000, China; 3Department of Food Science and Nutrition, College of Sciences, Taif University, P.O. Box 11099, Taif 21944, Saudi Arabia; 4Department of Mechanical Engineering, Faculty of Engineering, Taif University, P.O. Box 11099, Taif 21944, Saudi Arabia; mo.helal@tu.edu.sa; 5Department of Clinical Laboratory Sciences, College of Applied Medical Sciences, Taif University, P.O. Box 11099, Taif 21944, Saudi Arabia; khadigaah.aa@tu.edu.sa; 6Department of Physical Sport Science, College of Education, Princess Nourah bint Abdulrahman University, P.O. Box 84428, Riyadh 11671, Saudi Arabia; ahaljahani@pnu.edu.sa

**Keywords:** N-CDs, calcein, ratiometric fluorescent sensor, arginine, acetaminophen

## Abstract

In our study, a unique ratiometric fluorescent sensor for the rapid detection of arginine (Arg) and acetaminophen (AP) was constructed by the integration of blue fluorescent N-CDs and yellowish-green fluorescent calcein. The N-CD/calcein ratiometric fluorescent sensor exhibited dual emission at 435 and 519 nm under the same excitation wavelength of 370 nm, and caused potential Förster resonance energy transfer (FRET) from N-CDs to calcein. When detecting Arg, the blue fluorescence from the N-CDs of the N-CD/calcein sensor was quenched by the interaction of N-CDs and Arg. Then, the fluorescence of our sensor was recovered with the addition of AP, possibly due to the stronger association between AP and Arg, leading to the dissociation of Arg from N-CDs. Meanwhile, we observed an obvious fluorescence change from blue to green, then back to blue, when Arg and AP were added, exhibiting the “on–off–on” pattern. Next, we determined the detection limits of the N-CD/calcein sensor to Arg and AP, which were as low as 0.08 μM and 0.02 μM, respectively. Furthermore, we discovered that the fluorescence changes of the N-CD/calcein sensor were only responsible for Arg and AP. These results suggested its high sensitivity and specificity for Arg and AP detection. In addition, we have successfully achieved its application in bovine serum samples, indicating its practicality. Lastly, the logic gate was generated by the N-CD/calcein sensor and presented its good reversibility. Overall, we have demonstrated that our N-CD/calcein sensor is a powerful sensor to detect Arg and AP and that it has potential applications in biological analysis and imaging.

## 1. Introduction

As an α-amino acid that is used in the biosynthesis of proteins, arginine (Arg) is classified as a conditionally essential amino acid, especially in young mammals. It acts as an important synthetic precursor for protein, creatine, polyamine and nitric oxide [1,2], and is mainly involved in multiple physiological processes, including metabolism, immunomodulatory function, intestinal development, anti-tumor and anti-obesity pathways [3]. Arg deficiency in the human body could cause endothelium dysfunction, a late asthmatic response and difficulties in wound healing [4,5,6,7,8]. However, excessive Arg also leads to acute and chronic pancreatitis [9], increased perivascular NO levels [10], cellular damage [11] and even toxic effects [12].

Acetaminophen (AP) is a type of phenolic analgesic drug that exerts anti-inflammatory and antipyretic effects [13,14]. Nonetheless, overdose uptake and long-term use of AP could cause many medical problems, including asthma [15], renal injury [16], severe liver injury, coma or even death [17].

Therefore, it is important to monitor the concentrations of Arg and AP in the human body qualitatively and quantitatively to prevent potential serious diseases. To date, various methods have been exploited to detect Arg and AP, including high performance liquid chromatography(HPLC) [18], capillary zone electrophoresis(CZE) [19], amino acid analyzers [20], high-temperature paper chromatography [21], voltammetry [22], raman detected differential scanning calorimetry (RD-DSC)[23] and liquid chromatography-mass spectormetry(LC/MS) [24]. However, most of them are laborious, entail high consumption of material resources [25] and require professional operation. Therefore, it is highly necessary to develop a simple and easy approach to detect Arg and AP with high sensitivity and specificity.

Carbon dots (CDs), a new type of fluorescent carbon nanomaterial, have multiple advantages, including excellent optical properties, easy synthesis and modification, low toxicity and high stability [26]. Thus, CDs can be extensively used in several areas, such as drug transportation [27,28,29], biological imaging [30,31], cancer therapy [32], catalysis [33,34], sensing [35,36,37] and so on. In recent years, CDs have also been employed as a fluorescent sensor for detection and mostly exhibit a single-signal output. However, a single-signal output heavily depends upon the accurate measurement of the signal strength, and the background signals could easily disturb the signal strength and lead to potential fluctuations in outcomes, resulting in low accuracy. Therefore, ratiometric fluorescent sensors based on dual-signal output have been developed, and the detection of arginine or AP by ratiometric fluorescent sensors based on carbon dots has gradually gained more attention [38,39,40]. Compared with the single-signal output, the dual-signal output has the abilities of signal strength self-calibration, better signal-to-noise ratio, reduced background interference and superior accuracy and reliability [41,42,43,44,45,46].

In this study, we have designed a dual-emission ratiometric fluorescent sensor based on N-CDs and calcein for fluorescence and visual detection of Arg and AP, and the sensor has high sensitivity and selectivity. Briefly, our N-CD/calcein fluorescent sensor displays the “on–off–on” mode in sensing Arg and AP (Figure 1).

## 2. Experimental Part

### 2.1. Materials

Acetaminophen (AP), calcein, tryptophan (Trp), threonine (Thr), glucose (Glc), cysteine (Cys), phenylalanine (Phe), glycine (Gly), tyrosine (Tyr), urea (Urea), ascorbic acid (AA), serine (Ser) and other amino acids were purchased from Aladdin Chemical Reagent Co., Ltd. (Shanghai, China). NaCl, HCl, NaOH, quinine sulfate and other chemical reagents were obtained from Beijing Chemical Plant (Beijing, China). All chemicals used in the experiments were of analytically pure grade and no further refinement was required. Ultra-pure water generated from the ultra-pure water purification system was utilized in the experimental process.

### 2.2. Experimental Instruments

Transmission electron microscope (TEM) images were captured through the JEM-2100F electron microscope (JEOL, Tokyo, Japan). X-ray diffraction (XRD) analysis was performed by the D8 FOCUS (Bruker AXS, Karlsruhe, Germany). X-ray photoelectron spectroscopy (XPS) was performed by the ESCALAB250Xi X-ray photoelectron spectrometer (Thermo, Waltham, MA, USA). Fourier-transform infrared spectroscopy (FTIR) was performed by the NICOLET 380 FTIR spectrometer (Thermo, Waltham, MA, USA). The fluorescence spectra were acquired by the RF-5301PC fluorescence spectrophotometer (Shimadzu, Kyoto, Japan). The UV–vis absorption spectra were acquired on the TU-1901 UV–visible spectrophotometer (Purkinje, Beijing, China). Zeta potential was recorded on the Zetasizer Nano ZS90 (Malvern, UK). The exposure experiment was carried out using the WD-9403E UV analyzer (Liuyi, Beijing, China) with the power of 6 W.

### 2.3. Synthesis of N-CDs

First, 102.2 mg of tryptophan and 59.2 mg of threonine were mixed at a 1:1 ratio and dissolved in concentrated hydrochloric acid (pH = 2), stirring with a glass rod continuously. The mixed solution was then transferred to a 30 mL reaction kettle and heated at 180 °C for 8 h. After cooling down, the solution was filtered using a 0.22 μm filter membrane and further dialyzed through a dialysis bag (MWCO = 500 Da) for 2 days. Lastly, the dialyzed solution was freeze-dried in a freeze-dryer; then, we stored the N-CD solid powders in the refrigerator at 4 °C.

### 2.4. Quantum Yield of N-CDs

We performed quantum yield measurements, and used the below equation for calculation:$$Y_{x}=Y_{S}\times \frac{F_{X}}{F_{S}}\times \frac{A_{S}}{A_{X}}\times {\left(\frac{\eta_{S}}{\eta_{X}}\right)}^{2}$$

*Y* was the quantum yield, *F* was the integrated emission area, *A* was the absorbance and *η* was the refractive index. The subscript *x* and *s* refers to the measured substance and the standard substance, respectively.

### 2.5. Construction of N-CD/Calcein Ratiometric Fluorescent Sensor

Because the distribution ratio of N-CDs and calcein influenced the ability to detect Arg and AP, we calculated the N-CD/calcein sensor fluorescence intensity (ratio of F_1_/F_2_) under the same excitation wavelength of 370 nm and demonstrated the optimized ratio as 1:3. Briefly, we mixed 50 μL of 3.74 mg/mL N-CD solution with 150 μL of 5.61 mg/mL calcein solution and dissolved the mixture in 2 mL of ultra-pure water.

### 2.6. Stability Evaluation

We examined the stability of N-CDs, calcein and N-CD/calcein (50 μL, 3.74 mg/mL N-CDs; 150 μL, 5.61 mg/mL calcein) under various conditions, including pH (from 1 to 10), ionic strength (0, 0.2, 0.4, 0.6, 0.8, 1.0 mM) and illumination under a UV analyzer (0, 10, 20, 30, 40, 50, 60 min), and the excitation wavelength was 370 nm.

### 2.7. Characteristics of N-CD/Calcein Sensor in Arg and AP Detection

All fluorescence measurements and the fluorescence spectra acquisition were carried out at room temperature. For Arg detection, Arg at varied concentrations (ranging from 0 to 100 μM) was added into 2 mL of aqueous N-CD/calcein sensor. When detecting AP, a series of various concentrations of AP solutions (ranging from 0 to 3 μM) were added into the N-CD/calcein/Arg complex system (mixed with 100 μM Arg). The fluorescence color changes of our sensor were recorded by a camera under ultraviolet light at 365 nm. To determine its specificity, we examined the fluorescence response toward various substances including Na^+^, Mn^2+^, K^+^, Trp, Thr, Glc, Cys, Phe, Gly, Tyr, Urea and AA, and the concentrations were all 1 mM.

## 3. Results and Discussion

### 3.1. Characterization of N-CDs

In this study, we used tryptophan and threonine as the nitrogen source and hydrochloric acid as a co-reactant to synthesize the N-CDs by the one-pot hydrothermal method. High resolution transmission electron microscopy (HRTEM) image of N-CDs indicated that the CDs were approximately spherical (Figure 2a). The diameters of N-CDs were in the range of 2.0 to 5.0 nm, with an average value of 3.29 nm (Figure 2b). Furthermore, the crystal structure of the N-CDs showed only one obvious diffraction peak near 2θ = 23° [47] (Figure 2c), indicating their amorphous structure. The surface functional groups of the N-CDs were determined by FTIR. Figure 2d shows that multiple peaks were observed. The bonds from 3390 to 240 cm^−1^ corresponded to N−H, −OH and C−H, and 1650 cm^−1^ corresponded to C=O/C=C stretching vibration, respectively. Moreover, the peaks at 1450, 1327, 1110 and 744 cm^−1^ were ascribed to C−H bending, C−N stretching, C−O stretching and N−H out-of-plane bending vibration, respectively.

The chemical components and surface states of N-CDs were studied by XPS (Figure 3). The XPS full spectrum of N-CDs exhibited four obvious peaks at 197.62, 284.8, 400.04 and 531.77 eV corresponding to Cl2p, C1s, N1s and O1s, respectively (Figure 3a). The preliminary results suggested that our N-CDs were composed of Cl, C, N and O, and the atomic percentages of Cl, C, N and O were 1.44%, 70.01%, 8.61% and 19.94%, respectively. For each element’s high-resolution XPS spectrum, the C1s spectrum displayed four peaks at 283.78, 284.98, 286.18 and 288.68 eV, which corresponded to C−C/C=C, C−N, C−O and C=O [48,49], respectively (Figure 3b). Two fitted characteristic peaks at 398.18 and 399.98 eV were observed in the N1s spectrum, indicating the existence of N−H and C−N (Figure 3c) [50]. The O1s spectrum was decomposed into two peaks at 530.58 and 532.18 eV, which were ascribed to C−O and C=O [51], respectively (Figure 3d). As shown in the Cl2p diagram, the two peaks were attributed to metal chlorides, suggesting their irrelevance to elements of N-CDs (Figure 3e) [52]. Above all, the results indicated that the surfaces of the N-CDs had multiple functional groups, including amino, carboxyl and hydroxyl groups.

### 3.2. Optical Properties of N-CDs and Calcein

The fluorescence quantum yield of N-CDs measured by the reference method was as high as 68%, with an absorption peak at 370 nm (Figure 4a). Moreover, the λ_ex_ and λ_em_ of N-CDs were observed at 370 and 435 nm, respectively (Figure 4b). Interestingly, the N-CD solution was colorless under sunlight and presented obvious blue fluorescence under ultraviolet light (365 nm) (Figure 4b inset). This phenomenon was consistent with the color range shown in CIE color coordinates (0.15, 0.09) (Figure 4e).

For calcein, a characteristic absorption peak was shown at 495 nm (Figure 4c), while the λ_ex_ and λ_em_ were detected at 490 and 520 nm, respectively (Figure 4d). Furthermore, calcein presented a bright yellowish green fluorescence under a UV lamp (Figure 4d inset) and the CIE color coordinate was (0.20, 0.70) (Figure 4f).

### 3.3. Construction of N-CD/Calcein Ratiometric Fluorescent Sensor

To construct the ratiometric fluorescent sensor that could generate two emissions under a single excitation, we mixed N-CD and calcein solutions. Since the fluorescence response of the ratiometric fluorescent sensor depended on the distribution ratio of its components, the optimal proportion of N-CDs and calcein in the N-CD/calcein sensor was studied.

When the ratio of N-CDs and calcein was 1:1, the yellowish green fluorescence of calcein was mostly covered by the blue fluorescence of N-CDs (Figure 5) because the two emission peaks were not separated well. On the other hand, the calcein fluorescence was so strong that the N-CDs’ fluorescence was hidden when the N-CD/calcein ratio was changed to 1:4 and 1:5. Therefore, these results suggested that the change in fluorescence would not be obvious if the amount of calcein was too low or too high. Surprisingly, the two emission peaks were isolated well and their emitted fluorescence intensities at 435 and 519 nm were in good proportion when the N-CD/calcein ratio was 1:3. This indicated a more agile fluorescence color range, and the ratio of 1:3 was used in the follow-up experiments to ensure its best performance.

Fluorescence energy resonance transfer (FRET) was developed previously and is widely used in the biomedical field. FRET refers to the phenomenon that energy between two fluorophores is transmitted from the donor to the acceptor in a non-radiative manner through dipole–dipole coupling [53]. FRET should satisfy the following three requirements: (1) the donor can emit fluorescence, (2) there is a certain overlap between the emission spectrum of the donor and the absorption spectrum of the acceptor, (3) the acceptor is close enough to the donor, and the action distance is generally 2–8 nm.

Accordingly, under a single excitation wavelength at 370 nm, the N-CD/calcein sensor displayed two obvious emission signal peaks at 435 and 519 nm, corresponding to N-CDs and calcein, respectively (Figure 6a). Specifically, the emission peak fluorescence intensity of N-CDs was reduced, while it was increased for calcein, indicating that we successfully constructed the N-CD/calcein ratiometric fluorescent sensor. Given its self-correcting performance, our ratiometric fluorescent sensor was found to be more accurate and reliable than its single component by avoiding the inaccuracy caused by certain external factors during the measurement.

Next, we explored the possible mechanism of our N-CD/calcein sensor. The N-CDs had a strong emission at 435 nm, while the calcein had its maximum excitation at 490 nm (Figure 6b). A great overlap between the emission of N-CDs and the excitation of calcein could be observed. Then, the FRET parameters were calculated: the overlap integral (J) was 1.71 × 10^14^ nm^4^ M^−1^ cm^–1^; the energy transfer efficiency (E) was 0.13; the critical distance (R_0_) was 3.53 nm, and the binding distance between donor and acceptor (r_0_) was 4.85 nm. According to the above data, it was implied that the FRET might have occurred between them.

### 3.4. Characteristics of N-CD/Calcein Ratiometric Fluorescent Sensor

To investigate the characteristics of our N-CD/calcein sensor, we examined its performance under various conditions, including pH, ionic strength and illumination.

pH is one of the critical factors in actual detection and affects the fluorescence intensity of the sensor. Hence, we examined the appropriate pH for our sensor by the determination of pH effects on itself as well as on its components, N-CDs and calcein, individually. The fluorescence intensities of N-CDs and calcein were both elevated when the pH increased from 1 to 7 and reached their maximum strength at neutral pH at 7 (Figure 7). However, when the pH was greater than 7 and rose from 8 to 10, their intensities dramatically decreased. Meanwhile, we also calculated the ratio of the N-CD/calcein sensor (F_1_/F_2_). The results indicated that the ratio reached its lowest value at neutral pH at 7, and then increased when the pH was less than or greater than 7, suggesting that acidic and basic conditions both affected the performance of our N-CD/calcein sensor. Overall, we selected neutral pH at 7 for the subsequent experiments.

To examine the ionic strength effect of the fluorescence intensity, we explored its effects on N-CDs, calcein and the N-CD/calcein sensor (Figure 8). As shown in Figure 8, when the ionic strength was increased up to 1.0 mM, the fluorescence intensity of the N-CDs did not show a visible change, while the fluorescence intensity of calcein experienced a slight decrease at 0.8 and 1.0 mM. According to the calculation of the F_1_/F_2_ ratio, our N-CD/calcein sensor did not fluctuate; even the ionic strength was stable at 1.0 mM, indicating its tolerance to salinity.

Next, we sought to inspect the stability of our N-CD/calcein sensor. The intensity of N-CDs, calcein and the N-CD/calcein sensor remained stable for 60 min under ultraviolet light (365 nm) irradiation (Figure 9). This result suggested that our N-CD/calcein sensor overcame photobleaching and showed a benefit during analysis and testing over a long period of time.

Lastly, we examined the response time of the N-CD/calcein sensor to Arg and AP detection. With the existence of Arg, the fluorescence intensity started to decrease within 15 s and gradually dropped within 75 s; then, it remained steady until 120 s (Figure 10a). Instead, when AP was added to the N-CD/calcein/Arg complex system, the fluorescence intensity increased within 15 s and was elevated within 45 s, and it then remained constant until 90 s (Figure 10b). Therefore, 75 and 45 s were chosen for the subsequent experiments when detecting Arg and AP, respectively.

### 3.5. Ratiometric Fluorescence and Visual Detection of Arg and AP

To reveal the detailed information about Arg and AP detection, we examined the ratiometric fluorescence and color changes of our N-CD/calcein sensor under various concentrations of Arg and AP.

With the increasing concentration of Arg (C_Arg_), the fluorescence intensity of the N-CD/calcein sensor remained decreased and reduced to only 35% of its original intensity when the C_Arg_ reached 100 μM (Figure 11a). Specifically, the emission peak of N-CDs at 435 nm decreased more obviously than that of calcein at 519 nm when the C_Arg_ was increased. Meanwhile, the fluorescence color of the N-CD/calcein sensor changed from blue to green with the addition of 100 μM Arg (Figure 11a insets). By calculation, there was a strong linear relationship between the F_1_/F_2_ ratio (F_1_/F_2_ ratio = 1.73) and C_Arg_ (in the range of 0.1 to 100 μM) (Figure 11b). In detail, the calculated linear regression equation was F_1_/F_2_ = −0.00828C_Arg_ + 1.77789 (R^2^ = 0.9966), and the detection limit (LOD, S/N = 3) was 0.08 μM. Moreover, with an increasing amount of Arg added to the N-CD/calcein sensor, the fluorescence color changed gradually from blue to green (Figure 11b insets). Furthermore, we contrasted the performance of our sensor with previous studies for Arg detection and concluded that our sensor presented a wide linear range with low LOD (Table 1).

To evaluate the specificity of the N-CD/calcein sensor for Arg detection, we evaluated its performance for other substances, including three metal ions (Na^+^, Mn^2+^ and K^+^) and various organic small molecules (tryptophan (Trp), threonine (Thr), glucose (Glc), cysteine (Cys), phenylalanine (Phe), glycine (Gly), tyrosine (Tyr), urea (Urea) and ascorbic acid (AA)). Among all substances, the N-CD/calcein sensor exhibited the largest change in fluorescence intensity only with the existence of Arg, indicating its high selectivity for Arg detection (Figure 12a). Moreover, when Arg was mixed with interference substances (1 mM), Arg still dominated the fluorescence change of the N-CD/calcein sensor and the effect caused by the coexistence of the interfering substance could be neglected (Figure 12b). Above all, this information suggested that our N-CD/calcein sensor had high specificity to detect Arg.

Next, we started to examine the performance of our N-CD/calcein sensor for AP detection. Firstly, we generated the N-CD/calcein/Arg complex system by the addition of 100 μM of Arg. Then, we added different concentrations of AP (C_AP_) into the complex system and recorded the changes in the fluorescence intensity. As shown in Figure 13a, when increased C_AP_ was added, the fluorescence intensity of the N-CD/calcein/Arg complex system increased with the increased amount of AP and almost recovered to the original intensity when the concentration of AP reached 3 μM (Figure 13a). Briefly, the emission peak of N-CDs at 435 nm increased more obviously than that of calcein at 519 nm. As shown in the insets of Figure 13a, the mixed system displayed an obvious green color and then appeared blue after AP addition. Additionally, we also calculated the linear relationship between the C_AP_ and F_1_/F_2_ ratio (F_1_/F_2_ ratio = 0.79) and discovered a strong linear relationship when the concentrations of AP ranged from 0.03 to 3 μM (Figure 13b). In detail, the fitted linear equation was plotted as F_1_/F_2_ = 0.22967C_AP_ + 0.76752 (R^2^ = 0.9949) and the detection limit (LOD, S/N = 3) was 0.02 μM, indicating its sensitivity for AP detection. Simultaneously, when AP was added, the fluorescence color of the N-CD/calcein/Arg complex system changed gradually from green to blue (insets in Figure 13b). Moreover, we compared our N-CD/calcein/Arg complex system with previous studies and the comparison results are listed in Table 2.

Following AP detection, we next examined the performance of our N-CD/calcein/Arg complex system with various substances and found subtle fluorescence changes with the existence of other substances (Figure 14a). The quenched fluorescence was only recovered when AP was added into the complex system. When other substances (1 mM) were mixed with AP, their effects were inapparent, indicating the specificity of the N-CD/calcein/Arg complex system for AP detection (Figure 14b). Therefore, our N-CD/calcein/Arg complex system had high selectivity to AP.

### 3.6. Mechanism of the Ratiometric Sensing System

In this study, our N-CD/calcein ratiometric fluorescent sensor displayed the “on–off–on” pattern with the sequential addition of Arg and AP. Then, we explored the possible mechanism of the N-CD/calcein sensor for Arg and AP detection.

Due to the abundant hydroxyl and carboxyl on the surface, N-CDs had rich emission traps and the zeta potential analysis showed a negative potential (−11.3 mV) (Figure 15a) [62]. When Arg was mixed with our sensor, and the blue fluorescence of N-CDs from the N-CD/calcein sensor was quenched because Arg is a positively charged amino acid [63], and a guanidinium-carboxylate salt bridge was formed with the N-CDs based on the electrostatic interaction and hydrogen bond [64]. The structural change of N-CDs/Arg was demonstrated by FTIR spectrometry; compared to the FTIR spectrum of the N-CDs, the characteristic peak of carboxylate (1550 cm^−1^) appeared. The results further illustrated the formation of the guanidinium-carboxylate salt bridge (Figure 15b). Hence, it could be concluded that the reaction between carboxyl and guanidino groups was the main reason for the fluorescence quenching of N-CDs.

On the other hand, when AP was added into the N-CD/calcein/Arg complex system, as shown in Figure 15c, AP specifically incorporated with Arg through strongly interacting among O atoms from −COOH groups, N atoms from −NH_2_ and guanidine groups, forming the hydrogen and large unlocalized π bond [38]. Because of the stronger reactions between AP and Arg compared to those between Arg and the N-CDs, Arg was dissociated from the N-CD/calcein/Arg complex system and the blue fluorescence was recovered with the presence of AP.

### 3.7. Detection of Arg and AP in Bovine Serum

To assess the feasibility and practical application of the N-CD/calcein sensor, we decided to examine the Arg and AP content in bovine serum. The concentration of Arg was determined to be between 70.82 and 71.26 μM, which was consistent with previous reports. In addition, the concentration of Arg was re-calculated when three different concentrations (5, 15, 25 μM) of Arg were added into the bovine serum. As expected, the recovery rates were calculated to be as high as 95.5~104.4% (RSD < 0.7%) (Table 3). Furthermore, we also tested the ability of the N-CD/calcein/Arg complex system for AP detection. While the original AP content was not detected by the N-CD/calcein/Arg complex system, we were still able to detect the added AP and the recovery rates were measured to be within 99.3~102.0% (RSD < 2.7%) (Table 4). Altogether, these results indicated that our sensor has high practicability to detect Arg and AP in practical samples.

### 3.8. Logic Gate Application and Reversibility of N-CD/Calcein Sensor

As shown in Figure 16, we generated a fluorescent logic gate by considering Arg and AP as two inputs and the ratiometric fluorescence (F_1_/F_2_) as the output. The corresponding signs and the operation table of results were summarized (Figure 16a). Briefly, with regard to inputs, the existence of Arg and AP was defined as “1”, and absence was considered as “0”. As for the output, blue fluorescence indicated the “ON” state and was defined as “0”, while green fluorescence indicated the “OFF” state and was defined as “1”. When the inputs were in the (0, 0), (0, 1) and (1, 1) states, the sensor was “ON” and showed blue fluorescence. Instead, the sensor turned “OFF” and exhibited green fluorescence with only the existence of Arg when the input was in the (1, 0) state.

Reversibility is one of the most important characteristics for “on–off–on” fluorescent sensors. We added Arg and AP alternately to the N-CDs/calcein and recorded the intensity for estimating the reversibility of our sensor (Figure 17). Our N-CD/calcein sensor responded to the addition of Arg and AP within 10 cycles and the intensity changed regularly with each cycle, indicating its good reversibility.

## 4. Conclusions

In summary, we have prepared a novel ratiometric fluorescent (F_1_/F_2_) sensor through simply mixing N-CDs and calcein with possible FRET from N-CDs to calcein. The N-CD/calcein sensor was examined for the visual and ratiometric detection of Arg and AP, with the detection limits of 0.08 and 0.02 μM, respectively. Furthermore, we have proven that our N-CD/calcein sensor successfully examined the content of Arg and AP in bovine serum. In conclusion, our N-CD/calcein ratiometric fluorescent sensor could serve as a powerful tool for biological analysis and imaging in the future

## Figures and Tables

**Figure 1 nanomaterials-12-00976-f001:**
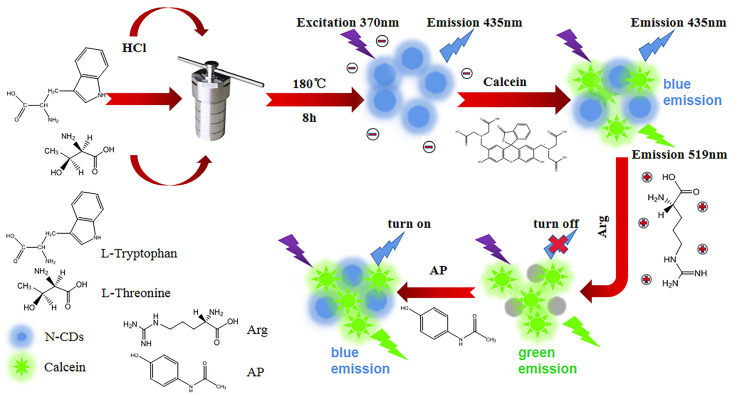
Schematic diagram of the construction of N-CD/calcein fluorescent sensor and its “on–off–on” mode for Arg and AP detection.

**Figure 2 nanomaterials-12-00976-f002:**
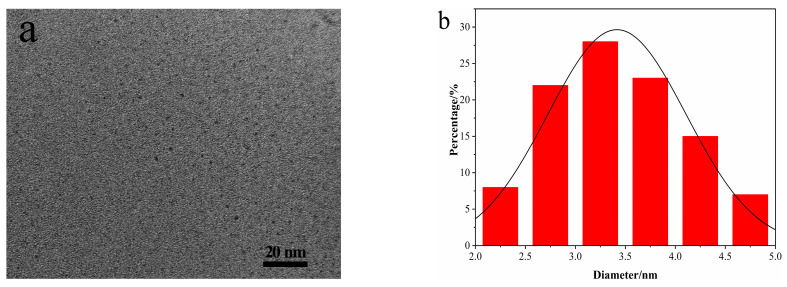
(**a**) HRTEM image of N-CDs; (**b**) particle size distribution of N-CDs; (**c**) XRD pattern of N-CDs; (**d**) FTIR spectrum of N-CDs.

**Figure 3 nanomaterials-12-00976-f003:**
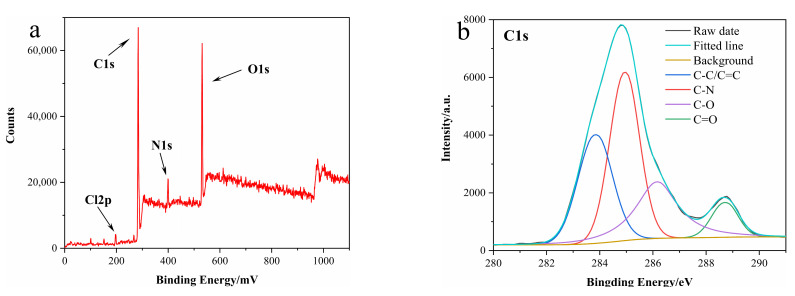
(**a**) XPS full spectrum; (**b**–**e**) high-resolution (**b**) C1s; (**c**) N1s; (**d**) O1s and (**e**) Cl2p XPS spectrum.

**Figure 4 nanomaterials-12-00976-f004:**
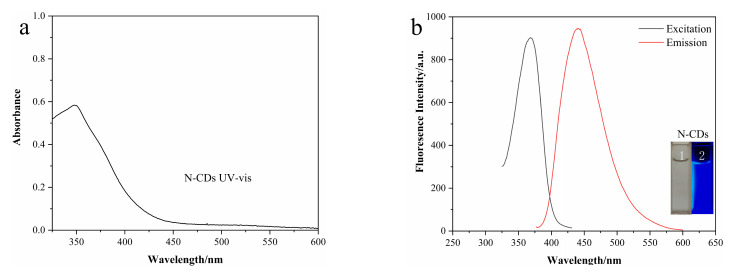
(**a**) UV–vis spectra of N-CDs; (**b**) fluorescence spectra of N-CDs; (**c**) UV–vis spectra of calcein; (**d**) fluorescence spectra of calcein; (**e**) calculated CIE coordinates according to the fluorescence spectrum of N-CDs; (**f**) calculated CIE coordinates according to the fluorescence spectrum of calcein. The insets show the corresponding fluorescence colors.

**Figure 5 nanomaterials-12-00976-f005:**
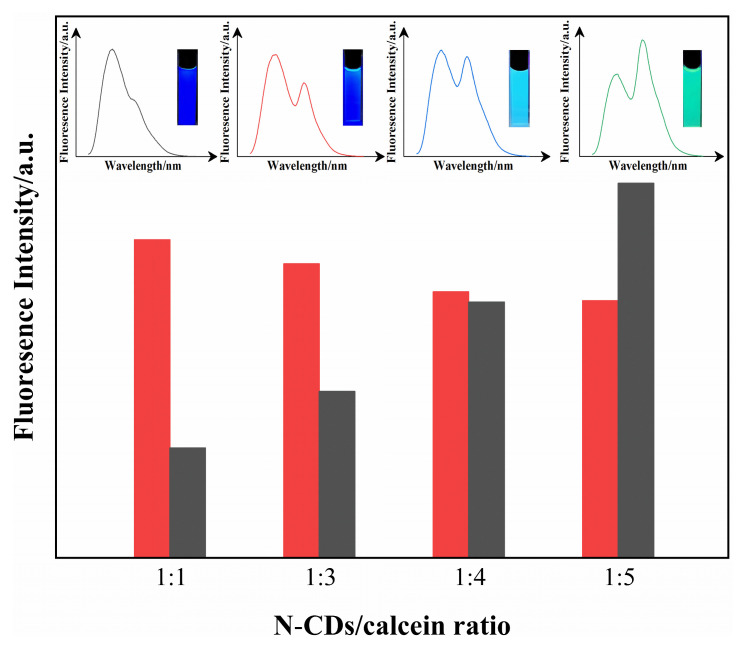
The corresponding fluorescence spectra and pictures under UV light (the inset) with different ratios of N-CDs and calcein in the N-CD/calcein mixture.

**Figure 6 nanomaterials-12-00976-f006:**
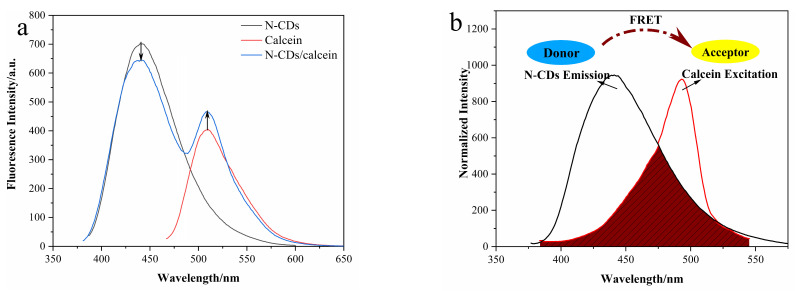
(**a**) FL spectra of N-CDs, calcein, N-CD/calcein (excited by 370 nm); (**b**) schematic diagram of possible FRET in N-CD/calcein sensor.

**Figure 7 nanomaterials-12-00976-f007:**
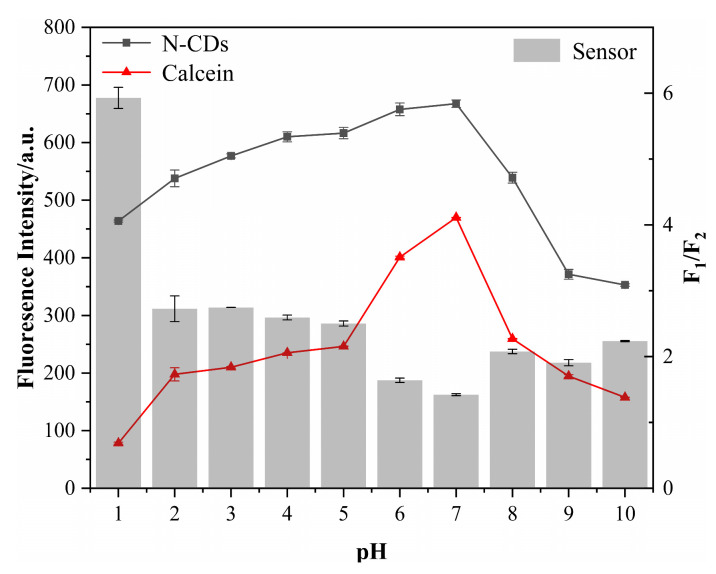
Effect of pH on the fluorescence intensity of N-CDs, calcein and N-CD/calcein sensor.

**Figure 8 nanomaterials-12-00976-f008:**
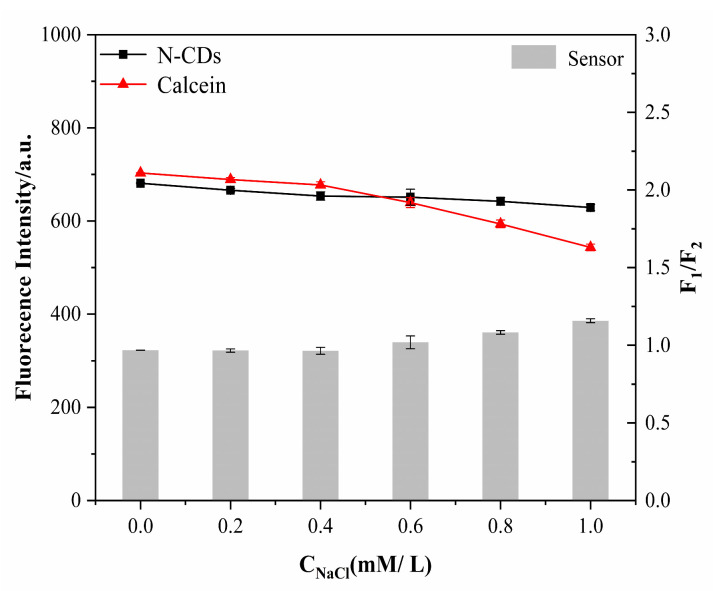
Effect of iconic strength on the fluorescence intensity of N-CDs, calcein and N-CD/calcein sensor.

**Figure 9 nanomaterials-12-00976-f009:**
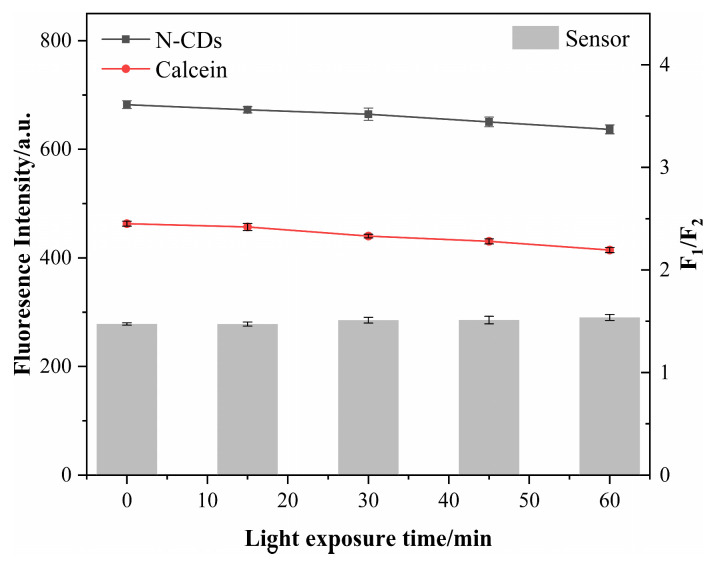
Effect of light exposure time on the fluorescence intensity of N-CDs, calcein and N-CD/calcein sensor.

**Figure 10 nanomaterials-12-00976-f010:**
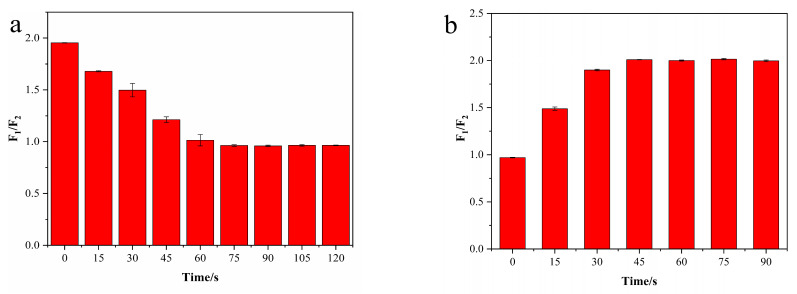
(**a**) The response time to Arg of our N-CD/calcein sensor; (**b**) the response time to AP of N-CD/calcein/Arg complex system.

**Figure 11 nanomaterials-12-00976-f011:**
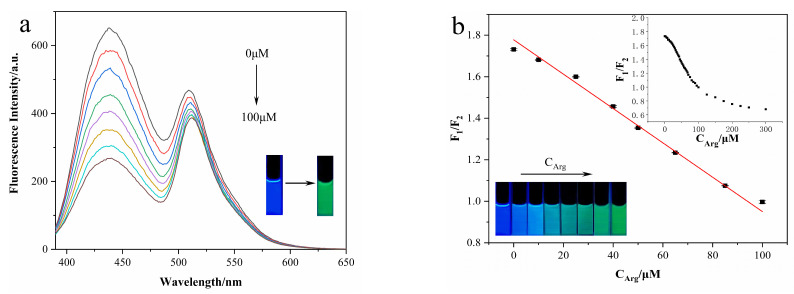
(**a**) Fluorescence intensity emission spectra of N-CD/calcein sensor with Arg addition. Insets show fluorescence color change of N-CD/calcein sensor in the absence (left) and presence (right) of the addition of 100 μM Arg. (**b**) Fitting curve between F_1_/F_2_ and C_Arg_. Insets show fluorescence color change of N-CD/calcein sensor with increased C_Arg_ (from left to right).

**Figure 12 nanomaterials-12-00976-f012:**
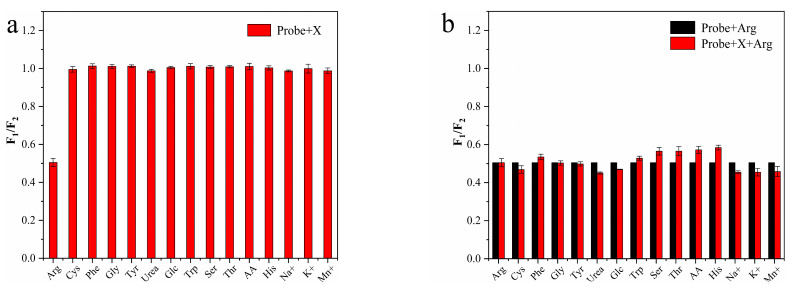
(**a**) F_1_/F_2_ ratio of N-CD/calcein sensor on Arg and other substances’ detection; (**b**) F_1_/F_2_ ratio of N-CD/calcein sensor on Arg detection with or without interference substances.

**Figure 13 nanomaterials-12-00976-f013:**
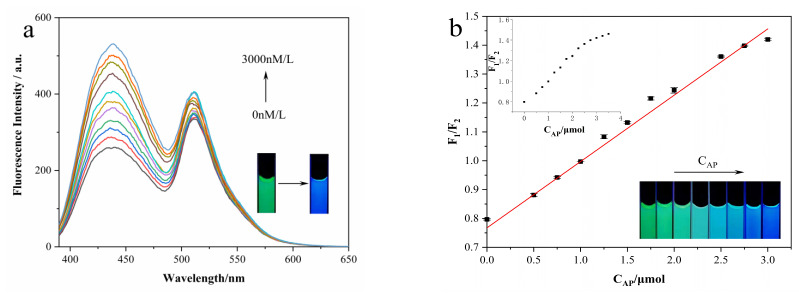
(**a**) Fluorescence intensity emission spectra of N-CD/calcein/Arg complex system with AP addition. Insets show fluorescence color change of N-CD/calcein/Arg complex system in absence (left) and presence (right) of the addition of 3 μM AP. (**b**) Fitting curve between F_1_/F_2_ and C_AP_. Insets show fluorescence color change of N-CD/calcein/Arg complex system with increased C_AP_ (from left to right).

**Figure 14 nanomaterials-12-00976-f014:**
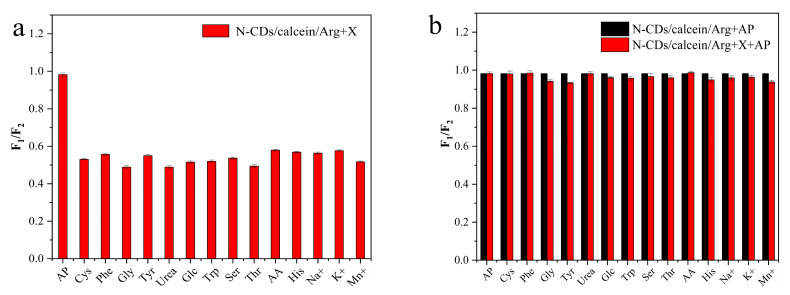
(**a**) F_1_/F_2_ ratio of N-CD/calcein/Arg complex system on AP and other substances’ detection; (**b**) F_1_/F_2_ ratio of N-CD/calcein/Arg complex system on AP detection with or without interference substances.

**Figure 15 nanomaterials-12-00976-f015:**
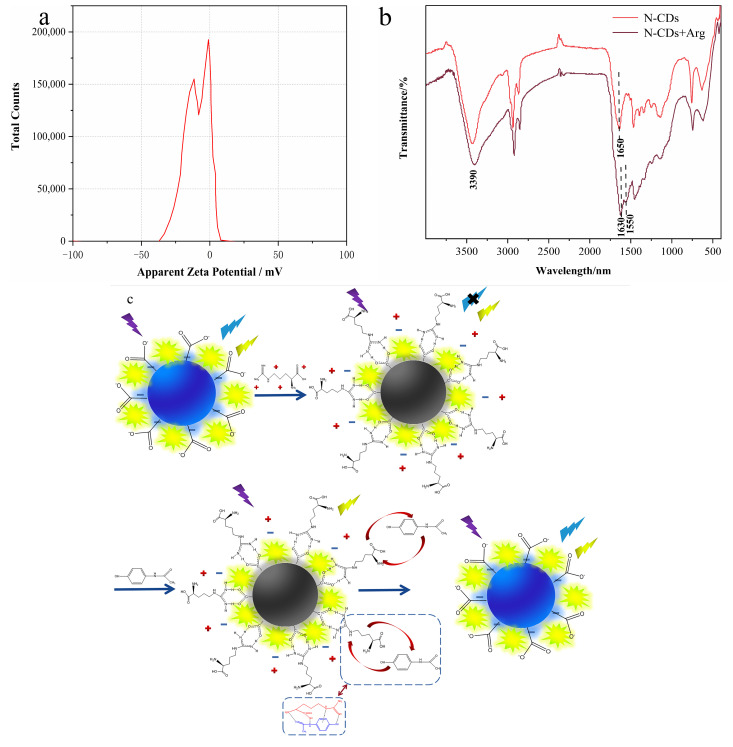
(**a**) Zeta potential of N-CDs; (**b**) FTIR spectra of N-CDs, N-CDs and Arg; (**c**) possible reaction mechanism of our N-CD/calcein sensor when detecting Arg and AP.

**Figure 16 nanomaterials-12-00976-f016:**
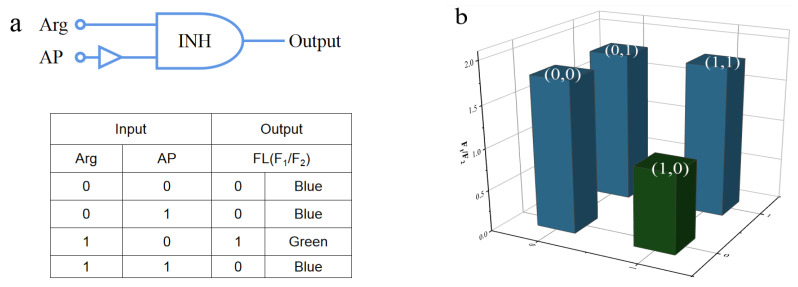
(**a**) Illustration of the fluorescent logic gates based on the N-CD/calcein sensor; (**b**) the histogram of logic gates with various inputs.

**Figure 17 nanomaterials-12-00976-f017:**
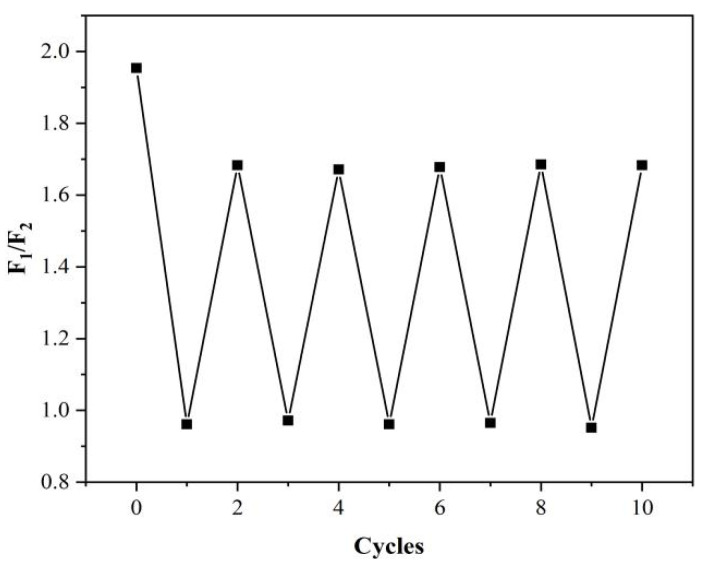
F_1_/F_2_ ratio of N-CD/calcein sensor with sequential addition of Arg and AP.

**Table 1 nanomaterials-12-00976-t001:** Comparison of different sensors for Arg sensing.

Sensor	Linear Range	LOD	Ref.
Mg-CDs	0.3–130 μM	0.15 μM	[25]
DNA-CuNC/CDs	0–100 μM	0.35 μM	[38]
Conductometric biosensor	−2.5–500 μM	−2.5 μM	[54]
AuNPs	1–6 μM	10 μM	[55]
Citrate-capped AuNPs	0.08–13.2 μM	0.02 μM	[56]
Capillary	30–1500 μM	10 μM	[57]
N-CDs/calcein	0.1–100 μM	0.08 μM	This work

**Table 2 nanomaterials-12-00976-t002:** Comparison of different sensors for AP sensing.

Sensor	Linear Range	LOD	Ref.
Ni_2_P nanosheets/GCE	0.5~4500 μM	0.11 μM	[58]
DNA-CuNC/CDs	0–100 μM	0.26 μM	[38]
Mo_2_C@BMZIFs/GCE	0.1–300 μM	0.03 μM	[59]
P-NC/GCE	3–110 μM	0.5 μM	[60]
ZnO-MoO_3_-C/GCE	2.5–2000 μM	1.14 μM	[61]
N-CDs/calcein/Arg	0.03–3 μM	0.02 μM	This Work

**Table 3 nanomaterials-12-00976-t003:** Determination of Arg in bovine serum.

Sample	Found in Sample (μM)	Added (μM)	Measured (μM)	Recovery (%)	RSD (%)
Serum	70.82	5	76.04	104.4	0.7
	72.04	15	86.37	95.5	0.2
	70.91	25	96.07	100.6	0.4

**Table 4 nanomaterials-12-00976-t004:** Determination of AP in bovine serum.

Sample	Found in Sample (μM)	Added (μM)	Measured (μM)	Recovery (%)	RSD (%)
Serum	0	0.5	0.51	102.0	0.8
	0	1.5	1.49	99.3	1.9
	0	2.5	2.49	99.6	2.7

## Data Availability

Available upon request from the corresponding author.

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
