# Peer review of "Construction of N-CDs and Calcein-Based Ratiometric Fluorescent Sensor for Rapid Detection of Arginine and Acetaminophen"

_nanomaterials, 2022, doi:10.3390/nano12060976_

Round 1

Reviewer 1 Report

Title: “Construction of N-CDs and Calcein Based Ratiometric Fluo-rescent Sensor for Rapid Detection of Arginine and Ace-taminophen”

Journal: Nanomaterials

Manuscript Number: nanomaterials-1623131

Haiyan Qi and colleagues have developed a ratiometric fluorescent sensor based on blue fluorescent N-CDs and yellowish-green fluorescent calcein to detect arginine and acetaminophen rapidly. Ratiometric fluorescent sensors based on N-CDs/calcein emit dual emissions at 435 nm and 519 nm under the same excitation wavelength at 370 nm, which led to förster resonance energy transfer (FRET) from N-CDs to calcein. In addition, they determined that our N-CDs/calcein sensor was capable of detecting Arg and AP as low as 0.08 μM and 0.02 μM, respectively. Furthermore, our N-CDs/calcein sensor has been demonstrated to be a powerful sensor for the detection of Arg and AP. However, I believe that the present paper lacks any novelty. Therefore, this manuscript should only be published after extensive and significant revisions. It is however necessary for the authors to devote a paragraph to highlight the novelty of the following publication.

There are a few points the author needs to address: 

Please read all the papers on N-doped carbon dots and sensor applications. More than a few hundred papers have been published. Would you please explain why your work is more valuable than others?

There is a need for the author to review the existing literature.

https://pubs.acs.org/doi/abs/10.1021/acs.jafc.9b03624

https://www.sciencedirect.com/science/article/abs/pii/S0003267021009892?via%3Dihub

https://pubs.acs.org/doi/abs/10.1021/acsami.7b08985

https://pubs.rsc.org/en/content/articlelanding/2016/tb/c6tb00519e/unauth  

I recommend that the Introduction, results, and conclusion be revised..

It is important to explain the process and to include the overall to förster resonance energy transfer (FRET) mechanism.

I am interested in learning the chemistry behind the sensor.

We expect the authors to provide high-resolution TEM images. Since the TEM image contains only a few particles, it is not possible to determine the size distribution from a few N CDs.

The authors are required to present high-resolution TEM images. Since it contains only a few particles, it is not possible to determine the size distribution from a few N CDs. 

The manuscript lacks any scientific insight. Therefore, you should provide additional information. 

Author Response

Response to the Reviewer1 comments:

  1. About “Please read all the papers on N-doped carbon dots and sensor applications. More than a few hundred papers have been published. Would you please explain why your work is more valuable than others?”

Our work has some key advantages, first the raw materials are amino acids, as they have good biocompatibility, innocuity. Nitrogen doped carbon dots are synthesized via one step hydrothermal method, experimental process is green. The high quantum yield is 68%, it is higher which is beneficial for the practical application. Second our fluorescent sensors based on dual-signal output, which has the abilities of signal strength self-calibration, better signal-to-noise ratio, reduced background interference and superior accuracy and reliability. So the proposed method obtains the most accurate. Third the tested substances are closely related to our life and has important implications.

  1. About “There is a need for the author to review the existing literature. https://pubs.acs.org/doi/abs/10.1021/acs.jafc.9b03624ï¼›https://www.sciencedirect.com/science/article/abs/pii/S0003267021009892?via%3Dihubï¼› https://pubs.acs.org/doi/abs/10.1021/acsami.7b08985ï¼›https://pubs.rsc.org/en/content/articlelanding/2016/tb/c6tb00519e/unauth”

We have added the literatures in the manuscript.

c.About “I recommend that the Introduction, results, and conclusion be revised.”

We have revised the introduction, results, and conclusion. We hope the revised version of the manuscript is satisfying.

d.About “It is important to explain the process and to include the overall to förster resonance energy transfer (FRET) mechanism.”

We have added the förster resonance energy transfer (FRET) mechanism in the manuscript.

e.About “I am interested in learning the chemistry behind the sensor.”

The chemistry mechanism behind the sensor is: when Arg is mixed with our sensor, the guanidinium-carboxylate salt bridge is formed between Arg and N-CDs, and when AP was added into the N-CDs/calcein/Arg complex system, AP specifically incorporated with Arg through strongly interacting among O atoms from -COOH groups, N atoms from -NH2 and guanidine groups, forming the hydrogen and big unlocalized π bond, because of stronger reactions between AP and Arg than Arg and N-CDs, Arg was dissociated from N-CDs/calcein/Arg complex system .

f.About “We expect the authors to provide high-resolution TEM images. Since the TEM image contains only a few particles, it is not possible to determine the size distribution from a few NCDs.”

We have restacked the TEM image, and we hope the picture quality is satisfying.

g.About “The manuscript lacks any scientific insight. Therefore, you should provide additional information. ”

In our work, there are some scientific insight and we also provide additional information. First we have designed a ratiometric fluorescent sensor which based on dual-signal output. And it has the abilities of signal strength self-calibration, better signal-to-noise ratio, reduced background interference and superior accuracy and reliability. Second the N-CDs/calcein ratiometric fluorescent sensor caused potential förster resonance energy transfer (FRET) from N-CDs to calcein and we have added the FRET parameters (FRET efficiency, spectral overlay integral, Forster distance and reaction mechanism to the manuscript, and further increase the scientific insight.

Reviewer 2 Report

In this study, blue fluorescent N-CDS and yellowish-green fluorescent calcein were combined to construct a unique ratiometric fluorescent sensor for rapid detection of arginine (ARG) and acetaminophen (AP). ARG is added to the fluorescence sensor, then AP is added, and the ratio of ARG to AP is obtained as the color changes. The "on-off-on" mode of the experimental design achieves the reversibility of the sensor. The dual-channel sensor design has better anti-interference ability. The experimental theory of this paper is established, the experimental data is complete, the organization is clear, the steps are clear, and it has good practicability. Here are some suggestions:

  1. The ‘on-off-on’ mode is displayed in Line 31, and it is speculated that the Line 30 should be ‘obvious fluorescence change from blue to green, back to blue’.
  2. Figure 1 shows the conversion of the "on-off-on" mode. It is recommended to indicate the main color, which can be more intuitive.
  3. It is recommended to place tables and instructions on the same page as much as possible.
  4. In Figure 2-b, the “Percentage/%” in vertical ordinate must place close to Figure 2-b, but not Figure 2-a. The same issue as in Figure 3-b, d; Figure 4-b,d; Figure 6-b; Figure 10-b; Fifure 12-b; Figure 14-b;
  5. The Figure 15 on Page 14 has too much white space under it, making it seem relatively empty.
  6. In Table 3, the Recovery=(Found—Added)/Found in sample, but in Table 4, the Recovery=Found/Added. Is the Recovery=(Found—Found in sample)/Added? Please reconfirm.

Author Response

Response to the Reviewer2 comments:

a.About “The ‘on-off-on’ mode is displayed in Line 31, and it is speculated that the Line 30 should be ‘obvious fluorescence change from blue to green, back to blue’.”

We have revised this sentence.

b.About “Figure 1 shows the conversion of the "on-off-on" mode. It is recommended to indicate the main color, which can be more intuitive.”

According to your suggestion, we have modified the picture and indicated the main color in the corresponding position.

c.About “It is recommended to place tables and instructions on the same page as much as possible.”

We have placed tables and instructions on the same page.

d.About “In Figure 2-b, the “Percentage/%” in vertical ordinate must place close to Figure 2-b, but not Figure 2-a. The same issue as in Figure 3-b, d; Figure 4-b,d; Figure 6-b; Figure 10-b; Fifure 12-b; Figure 14-b;”

We have revised the Figures in the manuscript.

e.About “The Figure 15 on Page 14 has too much white space under it, making it seem relatively empty.”

We have adjusted in the manuscript.

f.About “In Table 3, the Recovery=(Found—Added)/Found in sample, but in Table 4, the Recovery=Found/Added. Is the Recovery=(Found—Found in sample)/Added? Please reconfirm.”

Maybe we use the words in the tables and easily lead to misunderstanding, so we have changed the words. In table 3 and table 4, we use the same calculation formula: Recovery=(Measured-Found in sample)/Added. We have modified data in the tables

Reviewer 3 Report

The manuscript entitled, ‘Construction of N-CDs and Calcein Based Ratiometric Fluorescent Sensor for Rapid Detection of Arginine and Acetaminophen’ reported sysnthesis of carbon dots and its sensing applications. I am mentioning some loopholes of this work which should be accounted prior to publication;

  1. Why such precursor materials are taken is not clear.
  2. Did the author perform excitation dependent behavior of the CDs?
  3. How was the stability of the NCDs in different environments? Could the author enlighten about that?
  4. In Figure 4, it is shown that the NCDs are white light emitting. Could the author justify the CIE diagram?
  5. The NCDs synthesis mechanism and sensing could be emphasized with reporting some other’s published data. I am mentioning some of them; https://doi.org/10.1021/acsami.0c14527; https://doi.org/10.1016/j.foodchem.2020.128893; https://doi.org/10.1021/acs.langmuir.1c00471; https://doi.org/10.1016/j.foodchem.2022.132152.   
  6. Conclusions part should be modified with brief obtained results.

Author Response

Response to the Reviewer3 comments:

a.About “Why such precursor materials are taken is not clear.”

In our work, the raw materials are amino acids, as they have good biocompatibility, innocuity. Amino acids are green and environment-friendly, and are rich in nitrogen which can omit the process of dopping. And can gain carbon dots with high fluorescence quantum yield.

Considering single-signal output may be affected by the external environment and lead to potential fluctuations in outcome. In our work, we have designed a dual-emission ratiometric fluorescent sensor, so choose a suitable reference fluorophore is need.  Calcein has excellent and stable fluorescence properties, and emits yellowish-green light. And calcein has large Stokes Shift, and an obvious absorption peak between 200 nm and 400 nm. Therefore, the yellowish-green fluorescence can be displayed under the same excitation wavelength as the CDs. So we choose calcein as reference fluorophore.

b.About “Did the author perform excitation dependent behavior of the CDs?”

We have performed excitation wavelength dependence experiment. Because the emission wavelengths of CDs are not dependent on the excitation wavelengths, so we did not put it in the manuscript.

c.About “How was the stability of the N-CDs in different environments? Could the author enlighten about that?”

In our work, the N-CDs have excellent stability under high ionic strength and ultraviolet light (365 nm) irradiation. The fluorescence intensities of N-CDs elevated when pH increase from 1 to 7 and reach their maximum strength at neutral pH at 7. However, when pH is greater than 7 and go from 8 to 10, their intensities dramatically decrease. Our finding indicated that N-CDs could have significant potential for sensing applications in physiological environments at pH7.

In further, combine with the satisfactory results obtained in the experiment, we believe that the nitrogen-doped carbon dots synthesized by one-step hydrothermal method using amino acids as raw materials not only have high fluorescence quantum yield, but also have excellent fluorescence stability. If we use different kinds of amino acids with different structures as raw materials to synthesize CDs, these CDs may have some interesting qualities.

d.About “In Figure 4, it is shown that the NCDs are white light emitting. Could the author justify the CIE diagram?”

In Figure 4, it is shown the CDs presented obvious blue fluorescence under a 365 nm UV lamp, not white light emitting. The CIE coordinated according to the FL spectrum of N-CDs, so it is right. To clearly show the illuminant circs, we have adjusted point size in Figure 4 e, f.

e.about “The N-CDs synthesis mechanism and sensing could be emphasized with reporting some other’s published data. I am mentioning some of them; https://doi.org/10.1021/acsami.0c14527; https://doi.org/10.1016/j.foodchem.2020.128893; https://doi.org/10.1021/acs.langmuir.1c00471; https://doi.org/10.1016/j.foodchem.2022.132152.?”

We have added the literatures in the manuscript.

f.About “Conclusions part should be modified with brief obtained results.”

 We have modified the conclusions part. 

Reviewer 4 Report

The manuscript entitled " Construction of N-CDs and Calcein Based Ratiometric Fluorescent Sensor for Rapid Detection of Arginine and Acetaminophen” reports on the fabrication of a ratiometric dual emission sensor based on blue fluorescent N-carbon dots and yellow emissive calcein for the selective detection of arginine and acetaminophen substrates. In the work, the synthesis and characterization of N-CDs are reported, and the fabrication of the dual emissive sensor, when associated with calcein, ascribed to a FRET mechanism, under irradiation with 370 nm exciting light. The quenching of blue emission and changing in green fluorescence were exploited for revealing the Arg presence, while a recovery of the pristine blue emission of the sensor was observed with the concomitant presence of both Arg and AP. The developed sensor shows selectivity for the Arg and AP substrates and detection limits as low as 0.08 μM and 0.02 μM for Arg and AP, respectively. A possible sensing mechanism was proposed and the application of the sensor for detection of Arg and AP in bovine serum sample was also tested, together with a test concerning its reversibility. The scientific issues of the article sound appealing to the research interests, even if in terms of material and its application it is not new (a very similar work is reported in Microchimica Acta (2020) 187: 154), and the overall content is not always clear and full. In my opinion, to enhance the quality of the manuscript, Authors should address some improvements to their work. In general, I strongly recommend the Authors deeply revise the manuscript text, in order to improve the clarity and effectiveness and to remove eventual repetition of concepts.

Specifically:

  • The Introduction should be updated and expanded, including some recent works reporting the application of C-dots in the ratiometric sensing of Arg and advantages of dual emission sensors (see for example Microchimica Acta (2020) 187: 154, cited by the Authors, but I think it should be included in the Introduction; New J. Chem., 2019,43, 13234; J Mater Sci (2022) 57:576–588; etc…)
  • The experimental section is missing some experimental details, such as the quantum yield determination (i.e.reference standard used); specific experimental details concerning experiments to study the effect of pH and ionic strength; add details regarding the power of the UV radiation source used for the photobleaching experiment.
  • Among the multiple functional groups at the N-CD surface (amino, carboxyl, hydroxyl, pg. 5 lines 164-166) the only considered in the proposed sensing mechanism are the carboxylic groups. The Authors should add more information to this concern.
  • In the QY value, the use of the decimals has no sense. What is the uncertainty in the measurement, being the QY value obtained as an indirect measurement relatively to a reference standard?
  • Concerning the ratio of N-CDs and calcein (pag.7, line 205 and following): what kind of ratio (molar, weight) do the Authors refer to?
  • The choice of using 1:3 N-CDs calcein ratio is reported to be motivated by the good proportion and resolution between the two emission peaks of N-CDs and calcein. However, it appears that the 1:4 ratio also satisfies such conditions, being the relative intensities even more balanced.
  • Figure 5: the spectra and photos in the inset (upper panels) are too small. The figure is unreadable.
  • Concerning the proposed FRET mechanism between the N-CDs and calcein (pg. 8, lines 226 and following):
  • The Authors claim that the maximum of calcein excitation is at 470 nm. However, in both fig. 6b and 4d it doesn't seem to be like that at all, since excitation peaks in the figures lie at about 490 nm, almost 20 nm longer
  • Figure 6a reports the emission spectra of N-CDs, calcein, and their blend. Have the Authors an idea of the QY at 370 nm of the fluorophores? Indeed, especially for calcein, being the absorbance and excitation intensities at 370 nm very low, a very weak emission at that excitation wavelength is expected, also in the presence of the N-CDs.
  • Although a FRET process is proposed as a potential mechanism involving the N-CDs and calcein, such supposition remains a purely qualitative discussion. To validate a possible FRET from N-CDs to calcein, the FRET parameters (FRET efficiency, spectral overlay integral, Forster distance) should be calculated.
  • “pH is one of the critical factors in detection”. Moreover, the pH plays a key role also in the protonation/deprotonation equilibria of the functional groups pending at the CD surface. Therefore, what is the pH of the N-CDs aqueous dispersion? What occurs to the pH upon the mixing with the calcein? And how do the optical properties of both the NCDs and calcein change with respect to the pH variations? The spectral response, and not only the fluorescence intensity, can undergo large variations with changes in pH. And this can have an impact on the (FRET) mechanisms involving the two fluorophores (i.e. the FRET efficiency can sensibly change since variations in the spectral response can induce modification in the spectral overlap integral)
  • During the photobleaching experiments, do the absorption and emission spectra of the N-CDs and calcein change?
  • 10, lines 283-287 I suggest rephrasing such sentences, that are not clearly exposed. In general, for both the Arg and AP: from the emission spectra showed in fig. 11 a and 13 a, the wider reduction in fluorescence intensity of the band at 435 indicates the higher sensitivity of the blue emission to the Arg concentration, while there is not information of the quenching kinetics (“the emission peak of N-CDs at 435 nm decreased more rapidly than that of calcein..).It could be interesting to plot the emission intensities of the single fluorophores (N-CDs at 435, and calcein at 535) with respect to the Arg concentration, to evaluate the effect on the single color emission.
  • Figure 11b: the inset is really too small. The reader cannot draw any information from it. The standard deviation of the points in the main graph should be added. Please, consider the suggestion also for Fig. 13b
  • To evaluate the specificity of the sensor, the addition of several types of substances has been considered. However, details regarding i) the concentration of the added substances; ii) the initial F1/F2 ratio; iii) pH should be included (eventually in the experimental section). Please, apply the same observation in the case of the AP sensing.
  • The effect induced by the coexistent interference substances could, in general, be neglected (Fig. 12b). However, the F1/F2 ratio slightly increases if another amino acid is present in addition to Arg. What is the relative concentration of Arg/another amino acid? Does the sensor selectivity decrease more by increasing the concentration of the interference amino acid?
  • Finally, I have some problems (and doubts) regarding the section “Mechanism of the ratiometric sensing system”:
  • The protonation/deprotonation of all the functional groups pending at the N-CD surface is strictly dependent on the pH (I think that the sensing measurements have been performed at an almost neutral pH). What is the NCDs/calcein system charge at such pH? Moreover, why, in the proposed mechanism, only the carboxylic groups are considered to be present and to have a role in the reaction with Arg?
  • The reported zeta potential is -11.3 eV for N-CDs. Assuming that the presence of calcein in the mixture has no effect on the net charge, thus to consider the zeta potential measurements of only the N-CDs, in the plot of Fig. 15a, a double distribution of apparent zeta potential is shown: a peak at about -20 meV, and the more intense peak close to 0. I don’t understand the value reported in the text.
  • The formation of guanidinium carboxylate salt bridge can be supported by any experimental evidence? Do the pH, the spectral features, the zeta-potential distribution, the FT-IR spectra reveal some variations upon the addition of Arg, able to support such hypothesis?
  • There is a mistake on pg. 16, line 432 in the description of the inputs of the sensor. It should be (0, 0), (0, 1), and (1, 1) states for the blue emission instead of (0, 0), (1, 0), and (1, 1).

Author Response

Response to the Reviewer4 comments:

a.About “The Introduction should be updated and expanded, including some recent works reporting the application of C-dots in the ratiometric sensing of Arg and advantages of dual emission sensors (see for example Microchimica Acta (2020) 187: 154, cited by the Authors, but I think it should be included in the Introduction; New J. Chem., 2019,43, 13234; J Mater Sci (2022) 57:576–588; etc…).”

We have updated and expanded the introduction and added the recent works in the manuscript.

b.About “The experimental section is missing some experimental details, such as the quantum yield determination (i.e.reference standard used); specific experimental details concerning experiments to study the effect of pH and ionic strength; add details regarding the power of the UV radiation source used for the photobleaching experiment.”

We have added the experimental details in the experimental section.

c.About “Among the multiple functional groups at the N-CDs surface (amino, carboxyl, hydroxyl, pg. 5 lines 164-166) the only considered in the proposed sensing mechanism are the carboxylic groups. The Authors should add more information to this concern.”

There is guanidine group on the surface of arginine. And there are many functional groups at the N-CDs surface (amino, carboxyl, hydroxyl), but only the carboxyl is the dominant functional group because that the guanidinium-carboxylate salt bridge can be formed between guanidine and carboxyl. Further we add infrared spectrum of N-CDs+Arg in the manuscript, compared with the N-CDs spectrum, the carboxylate (1550cm−1) peak appeared, the result confirm the formation of guanidinium-carboxylate salt bridge. Please see manuscript for detailed description.

d.About “In the QY value, the use of the decimals has no sense. What is the uncertainty in the measurement, being the QY value obtained as an indirect measurement relatively to a reference standard?”

We use the reference method to determine the quantum yield of CDs according to the formula mentioned in 2.4 section. Finally, according to your suggestion, the QY value change to 68%.

e.About “Concerning the ratio of N-CDs and calcein (pag.7, line 205 and following): what kind of ratio (molar, weight) do the Authors refer to?”

In the 2.5 part have given the mass concentrations of N-CDs and calcein, then we mixed them by the different volumes. And in the whole paper, we all use the same way. When the concentration is determined, the mass is directly proportional to volume. So the radio is mass ratio.

f.About “The choice of using 1:3 N-CDs calcein ratio is reported to be motivated by the good proportion and resolution between the two emission peaks of N-CDs and calcein. However, it appears that the 1:4 ratio also satisfies such conditions, being the relative intensities even more balanced.”

When the N-CDs/calcein ratio is 1:4, the relative intensities more balanced. But in terms of colour, the 1:4 ratio when the N-CDs/calcein ratio is 1:4 the yellow-green fluorescence of calcein was too strong where mostly masked the blue fluorescence of N-CDs, the color change would not be obvious in subsequent experiments. So we choose 1:3.

g.About “Figure 5: the spectra and photos in the inset (upper panels) are too small. The figure is unreadable.”

We have adjust the size of picture.

h.About “     Concerning the proposed FRET mechanism between the N-CDs and calcein (pg. 8, lines 226 and following): The Authors claim that the maximum of calcein excitation is at 470 nm. However, in both fig. 6b and 4d it doesn't seem to be like that at all, since excitation peaks in the figures lie at about 490 nm, almost 20 nm longer”

This is a mistake, the excitation wavelength is 490 nm, we have changed.

i.About “      Figure 6a reports the emission spectra of N-CDs, calcein, and their blend. Have the Authors an idea of the QY at 370 nm of the fluorophores? Indeed, especially for calcein, being the absorbance and excitation intensities at 370 nm very low, a very weak emission at that excitation wavelength is expected, also in the presence of the N-CDs.”

In fact, the λex of CDs is 370 nm, so the the emission peak is strong. Calcein has large Stokes Shift, and an obvious absorption peak between 200 nm and 400 nm. When the calcein is excited at 370, the yellowish-green fluorescence can be displayed and the emission intensities is also strong. The results can be seen in Figure 6a.

j.About “Although a FRET process is proposed as a potential mechanism involving the N-CDs and calcein, such supposition remains a purely qualitative discussion. To validate a possible FRET from N-CDs to calcein, the FRET parameters (FRET efficiency, spectral overlay integral, Forster distance) should be calculated.”

We have calculated the FRET parameters (FRET efficiency, spectral overlay integral, Forster distance, added the values in the manuscript.

k.About “pH is one of the critical factors in detection”. Moreover, the pH plays a key role also in the protonation/deprotonation equilibria of the functional groups pending at the CD surface. Therefore, what is the pH of the N-CDs aqueous dispersion? What occurs to the pH upon the mixing with the calcein? And how do the optical properties of both the NCDs and calcein change with respect to the pH variations? The spectral response, and not only the fluorescence intensity, can undergo large variations with changes in pH. And this can have an impact on the (FRET) mechanisms involving the two fluorophores (i.e. the FRET efficiency can sensibly change since variations in the spectral response can induce modification in the spectral overlap integral)”

The pH of the N-CDs aqueous dispersion is about 6.7, when mixing with the calcein, the pH is about 7.0, which remains largely the same.

The emission wavelengths of both the NCDs and calcein are not change with respect to the pH variations. So pH is not effect the FRET efficiency and the spectral overlap integral

l.About “During the photobleaching experiments, do the absorption and emission spectra of the N-CDs and calcein change?”

During the photobleaching experiments, the absorption and emission spectra of the N-CDs and calcein are not change.

m.About “    10, lines 283-287 I suggest rephrasing such sentences, that are not clearly exposed. In general, for both the Arg and AP: from the emission spectra showed in fig. 11 a and 13 a, the wider reduction in fluorescence intensity of the band at 435 indicates the higher sensitivity of the blue emission to the Arg concentration, while there is not information of the quenching kinetics (“the emission peak of N-CDs at 435 nm decreased more rapidly than that of calcein..). It could be interesting to plot the emission intensities of the single fluorophores (N-CDs at 435, and calcein at 535) with respect to the Arg concentration, to evaluate the effect on the single color emission.”

According to your suggestion, we have modified the corresponding sentence in the manuscript.

We have studied the effect on the single color emission in the experimental process. When the Arg add, the emission intensity of N-CDs (435 nm) decrease more obviously compare with the emission intensity of calcein (535 nm) is  basically unchanged. Both the emission wavelengths of them are not dependent on the excitation wavelengths, and we did not put them in the manuscript.

n.About “     Figure 11b: the inset is really too small. The reader cannot draw any information from it. The standard deviation of the points in the main graph should be added. Please, consider the suggestion also for Fig. 13b”

We have adjusted the Figure 11b.

The standard deviation of the points have been added in the original manuscript, due to the standard deviation is little, so it's not obvious.

o.About “To evaluate the specificity of the sensor, the addition of several types of substances has been considered. However, details regarding i) the concentration of the added substances; ii) the initial F1/F2 ratio; iii) pH should be included (eventually in the experimental section). Please, apply the same observation in the case of the AP sensing”

We have added all the details which you mentioned.

p.About “The effect induced by the coexistent interference substances could, in general, be neglected (Fig. 12b). However, the F1/F2 ratio slightly increases if another amino acid is present in addition to Arg. What is the relative concentration of Arg/another amino acid? Does the sensor selectivity decrease more by increasing the concentration of the interference amino acid?”

The competition substance concentration is 1 mM, it is ten times the concentration of measured substance.

In Fig. 12b, When the another amino acid are mixed with Arg, in the N-CDs/calcein system, the fluorescence increase  changes very little, this result suggesting the specificity of the system to Arg detection. So the sensor selectivity is not decrease by increasing the concentration of the interference amino acid.

q.About “The protonation/deprotonation of all the functional groups pending at the N-CD surface is strictly dependent on the pH (I think that the sensing measurements have been performed at an almost neutral pH). What is the NCDs/calcein system charge at such pH? Moreover, why, in the proposed mechanism, only the carboxylic groups are considered to be present and to have a role in the reaction with Arg?”

At pH7, the charge of N-CDs/calcein system is -14.8 eV.

There is no obvious reaction between CDs and other detected substances, but there is an obvious reaction between CDs and arginine. Because the guanidinium-carboxylate salt bridge can be formed between guanidine and carboxylic groups. And the FT-IR can prove this conclusion. Please see manuscript for detailed description.

r.About “The reported zeta potential is -11.3 eV for N-CDs. Assuming that the presence of calcein in the mixture has no effect on the net charge, thus to consider the zeta potential measurements of only the N-CDs, in the plot of Fig. 15a, a double distribution of apparent zeta potential is shown: a peak at about -20 meV, and the more intense peak close to 0. I don’t understand the value reported in the text.”

There are two peaks one at about -20 meV, and the more intense peak close to 0. The value of zeta potential for N-CDs(11.3 eV) is calculated by using the following equations: .

s.About “The formation of guanidinium carboxylate salt bridge can be supported by any experimental evidence? Do the pH, the spectral features, the zeta-potential distribution, the FT-IR spectra reveal some variations upon the addition of Arg, able to support such hypothesis?”

The FT-IR spectra reveal some variations upon the addition of Arg, able to support formation of guanidinium carboxylate salt bridge. we add infrared spectrum of N-CDs+Arg in the manuscript, compared with the N-CDs spectrum, the carboxylate (1550cm−1) peak appeared, the result confirm the formation of guanidinium-carboxylate salt bridge. Please see manuscript for detailed description.

t.About “There is a mistake on pg. 16, line 432 in the description of the inputs of the sensor. It should be (0, 0), (0, 1), and (1, 1) states for the blue emission instead of (0, 0), (1, 0), and (1, 1).”

We have corrected the mistake.

Round 2

Reviewer 1 Report

Accept

Author Response

Response to the Reviewer1 comments:

Accept

Thanks a lot

Reviewer 3 Report

Now it can be accepted in its present form

Author Response

Response to the Reviewer3 comments:

Now it can be accepted in its present form

Thanks a lot

Reviewer 4 Report

The Authors revised the manuscript according to with my suggestions but, in my opinion, some issues still need to be addressed before the contribution could be accepted for publication. In particular, I suggest accurately revising the English language and punctuation. Personally, I appreciated the effort of the Authors to provide a timely response to the Reviewers' comments. But, probably due to the short time available for the response, often the improvements to the manuscript were not of adequate precision and attention for high-quality work for publication in Nanomaterials.

Some not exhaustive examples:

Line 105 --- exposure experiment instead of exposure time

Line 122 --- A dot is missing after “solvent”

Line 209---(b) FL spectra… What does FL mean?

Figure 5: Axis labels in the upper panels still remain too small

The caption of Figure 5: The fluorescence intensities and corresponding fluorescence colors… It should be Inset: corresponding fluorescence spectra and pictures under UV light ….

More relevant comments:

  1. Figure 2a shows a TEM image with higher magnification with respect to the previous version of the Figure, as the HR-TEM allows much higher resolution performance.
  2. Lines 232-240. FRET is not a technique, please revise this paragraph, there are also other spelling errors and punctuation missing.
  3. Please, check the spectral overlap integral value and units. In my opinion, it should be better to use the units of nm or cm for the wavelength, and units of M–1 cm–1 for the extinction coefficient, as recommended by one of the leading experts in the field, J. R. Lakowicz, in Principles of Fluorescence Spectroscopy, Springer Ed.

In light of the FRET efficiency value found, which is quite low (0.13), the mechanism of the N-CD /calcein sensor should be reviewed or commented differently, since it is really difficult to explain the good working of the sensor with such limited FRET performance.

Author Response

Response to the Reviewer4 comments:

  1. About “Line 105 --- exposure experiment instead of exposure time”

We have revised this sentence.

b.About “Line 122 --- A dot is missing after “solvent””

We have added the dot.

c.About “Line 209---(b) FL spectra… What does FL mean?”

The FL spectra means fluorescence spectra and we have change the caption.

d.About “Figure 5: Axis labels in the upper panels still remain too small. The caption of Figure 5: The fluorescence intensities and corresponding fluorescence colors… It should be Inset: corresponding fluorescence spectra and pictures under UV light …. ”

We have revised the Figures in the manuscript.

e.About “Figure 2a shows a TEM image with higher magnification with respect to the previous version of the Figure, as the HR-TEM allows much higher resolution performance.”

In order to more articulate problem, we have change the size of the ruler.

f.About “Lines 232-240. FRET is not a technique, please revise this paragraph, there are also other spelling errors and punctuation missing.”

We have revised this paragraph in the manuscript.

g.About “Please, check the spectral overlap integral value and units. In my opinion, it should be better to use the units of nm or cm for the wavelength, and units of M–1 cm–1 for the extinction coefficient, as recommended by one of the leading experts in the field, J. R. Lakowicz, in Principles of Fluorescence Spectroscopy, Springer Ed.”

We have modified them.

h.About “In light of the FRET efficiency value found, which is quite low (0.13), the mechanism of the N-CD /calcein sensor should be reviewed or commented differently, since it is really difficult to explain the good working of the sensor with such limited FRET performance.”

PET (Photo‑induced electron transfer), ICT (Intramolecular charge transfer) and FRET are the most commonly used to explain the mechanism of signal transmission for designing ratiometric fluorescent sensors. In our paper, we excluded the PET and ICT response mechanism mechanism. And we have obtained the relevant parameters of fluorescence resonance energy transfer. Finally, the above phenomena successfully prove the construction of ratiometric fluorescent probe.

Although the FRET efficiency value is 0.13, but it does not affect the detection mechanism and the linear relationship between F1/F2 ratio and concentration of detected substance. So it does not affect the properties of the ratiometric fluorescent sensors.
